# *N*-3 Fatty Acids in Seafood Influence the Association Between the Composite Dietary Antioxidant Index and Depression: A Community-Based Prospective Cohort Study

**DOI:** 10.3390/antiox13111413

**Published:** 2024-11-18

**Authors:** Junhwi Moon, Minji Kim, Yangha Kim

**Affiliations:** 1Department of Nutritional Science and Food Management, Ewha Womans University, Seoul 03670, Republic of Korea; jhm218@naver.com (J.M.); kbkjmmj2012@naver.com (M.K.); 2Graduate Program in System Health Science and Engineering, Ewha Womans University, Seoul 03760, Republic of Korea

**Keywords:** seafood, *n*-3 polyunsaturated fatty acids, composite dietary antioxidant index, depression, longitudinal study

## Abstract

Accumulating evidence suggests that seafood and its components, such as eicosapentaenoic acid (EPA) and docosahexaenoic acid (DHA), are associated with mental health. However, little is known regarding whether the status of *n*-3 polyunsaturated fatty acids (PUFAs) modify the effect of dietary antioxidants on depression. The main purpose of study is to investigate longitudinal associations between seafood consumption and depression among 2564 participants aged 40–69 years using data from the Korean Genome and Epidemiology Study. The composite dietary antioxidant index (CDAI) and dietary intake were measured by a validated 106-item food frequency questionnaire and depression was assessed using the Beck Depression Inventory (BDI). The Cox’s proportional hazard model was used to examine the risk of depression according to seafood consumption. During an 8-year follow-up period, 165 (11.9%) men and 224 (18.9%) women experienced depression. After adjustment for confounders, the risk of depression was inversely associated with seafood consumption, with a 42% lower risk (HR _T5 vs. T1_ = 0.58, 95% CI: 0.35–0.98, *p* = 0.040) only being found among women. In a group with a high *n*-3 PUFA intake, CDAI scores were negatively correlated with BDI scores (r = −0.146, *p* < 0.001) among women. Seafood consumption might lead to more favorable outcomes against depression if accompanied by an increased intake of foods that are rich in antioxidants.

## 1. Introduction

Depression is a common mental disorder that affects mood, behavior and overall health [1]. The global prevalence of depressive disorders was estimated to be around 3.44% in 2017, which increased sharply during the COVID-19 outbreak. The recent meta-analysis has suggested that the proportion of depression in the general population might be seven times higher over this period, accounting for 25% [2]. According to the National Mental Health Survey of Korea 2021, the lifetime prevalence of having any mental disorders among Korean adults was 27.8%, indicating that about one in four Korean adults experienced a psychiatric disorder during their lifetime [3]. Therefore, there is an increasing interest in how modifiable lifestyle factors, such as smoking, physical activity, alcohol consumption, and diet, may affect mental health [4,5].

A growing number of studies have investigated the association between seafood consumption and the risk of depression. A meta-analysis of 10 prospective cohort studies revealed that an increment of one serving/week of fish was associated with an 11% lower risk of depression [6]. In a prospective study of Japanese adult employees, increased intake of seaweed was associated with a decreased incidence of depressive symptoms at the 3-year follow-up [7]. Regarding dietary patterns, a meta-analysis of nine cross-sectional studies showed that greater adherence to the Mediterranean diet, characterized by a high intake of olive oil, plant products, fish and seafood, was associated with a 28% lower risk of depression [8]. However, no association was observed between fish consumption and depression in some cross-sectional studies [9,10,11].

*N*-3 polyunsaturated fatty acids (PUFAs), such as eicosapentaenoic acid (EPA) and docosahexaenoic acid (DHA), are components of seafood that are considered to be beneficial for mental health [12]. Accumulated evidence from several studies has suggested the role of *n*-3 PUFA on depression through potential mechanisms involving neuroendocrine modulation and the regulation of inflammatory statuses [13].

The composite dietary antioxidant index (CDAI) is a reliable nutritional tool that assesses individual antioxidant intake profiles [14]. In the Shanghai Women’s Health Study, the Dietary Antioxidant Quality Score and CDAI are highly correlated and are inversely associated with inflammatory mediators such as interleukin-1β and tumor necrosis factor-alpha (TNF-α) [15]. Increased levels of these proinflammatory cytokines may contribute to the development of depression [13]. Taking into consideration the potential anti-inflammatory effects of *n*-3 PUFAs, it can be speculated that *n*-3 PUFA, in combination with various antioxidants, might lead to more favorable outcomes against depression. However, to our knowledge, few studies have addressed the related issues [16,17].

In the current study, we aimed to examine longitudinal associations between seafood consumption and depression using data from the Korean Genome and Epidemiology Study (KoGES), a large community-based cohort study. We also investigated whether *n*-3 PUFA status modifies the effect of dietary antioxidants on depression.

## 2. Materials and Methods

### 2.1. Study Population

The data used in this study were obtained from a community-based Ansan–Ansung cohort study, part of the KoGES, to identify risk factors for chronic disease among the general Korean population. Detailed information on the study has been described elsewhere [18]. Briefly, 10,030 Korean adults aged 40–69 years who lived in Ansan (urban) and Ansung (rural) were recruited at baseline between 2001 and 2002, and follow-up examinations were conducted biennially. The second follow-up examination provided information on the depression levels, so we used these data as the baseline. Data from the baseline (2005–2006) through the sixth examination (2013–2014) were used for the current study. Among the 7515 participants, we excluded participants aged 65 and over for the purpose of focusing on middle-aged Koreans (*n* = 1740). Participants who reported implausible total energy intakes (<500 kcal/day or >4000 kcal/day, *n* = 80), those who never participated in the follow-up examinations (*n* = 164), those with missing information on covariates (*n* = 16), and those who did not respond to the depression questionnaire at baseline (*n* = 2577) were excluded. Additionally, 374 participants with depression at baseline were excluded. Therefore, 2564 participants (1379 men and 1185 women) were included in the final analysis (Figure 1).

### 2.2. Ethical Approval

This study was conducted according to the guidelines laid down in the Declaration of Helsinki, and all procedures involving human subjects were approved by the Institutional Review Board of Ewha Womans University (No. 202208-0010-01). Written informed consent was obtained from all individuals who participated in the study.

### 2.3. Assessment of Depression

At baseline, the level of depression in participants was assessed using the Beck Depression Inventory (BDI) developed by Beck et al. [19]. The validity of the BDI has been reported elsewhere [20]. The BDI consists of 21 items, including cognitive, emotional, motivational, and physical symptoms [19,21]. Each item ranges from ‘strongly disagree’ (0) to ‘strongly agree’ (3) based on a 4-point Likert scale. The total BDI score is calculated by summing the scores of each subscale. A higher BDI score indicates a higher level of depression. Depression was defined as a total BDI score ≥16, in accordance with previous studies of the Korean population [22,23].

### 2.4. Assessment of Food Consumption and Nutrients Intake

Dietary data were collected using a 106-item semi-quantitative food frequency questionnaire (SQFFQ) developed for Korean adults. The validity and reproducibility of the SQFFQ have been described elsewhere [24]. The food items listed in the SQFFQ were categorized into 13 groups based on the previous study [25] (Appendix A).

Food consumption was assessed at baseline of the study, concerning the participant’s dietary intake over the past year. Participants were asked to provide their average food frequency (on a 9-point scale of ‘almost none’, ‘once a month’, ‘twice or three times a month’, ‘once or twice a week’, ‘twice or three times a week’, ‘five or six times a week’, ‘once a day’, ‘twice a day’, and ‘three times a day’) and the average portion size (on a 3-point scale of ‘0.5 times the reference’, ‘reference’, and ‘1.5–2.0 times the reference’) for each food item. The duration of the seasonal variety of fruit consumption was divided into 4 categories (on a 4-point scale of 3, 6, 9, and 12 months).

To estimate seafood consumption, which comprised fish, shellfish, and seaweed, we multiplied the reported intake frequency of each food item in the SQFFQ by the reported portion size. Participants were divided into 5 groups according to quintiles of seafood consumption. The 10th revision of the Korean food composition table (KFCT), updated every five years by the Rural Development Administration [26], was used to evaluate the daily intakes of *n*-3 PUFAs. Daily nutrient intakes and calories were calculated from the food intake measured by the SQFFQ using the computer-aided nutritional analysis program (CAN Pro), a nutrient database developed by the Korean Nutrition Society [27]. The participants were divided into 2 groups based on the median value of intakes regarding the effects of *n*-3 PUFAs status.

### 2.5. Assessment of Composite Dietary Antioxidant Index

The CDAI was calculated using a modified version developed by Wright et al. [28]. The CDAI was the sum of six dietary minerals and vitamins (manganese, selenium, zinc, vitamins A, C, and E), and the daily intakes were evaluated based on food consumption through the use of the KFCT. The calculation formula was as follows:$$CDAI=\sum_{i=1}^{n=6}\frac{Individual\;Intake-Mean}{SD}$$

### 2.6. Data Collection

All participants were interviewed about their sociodemographic and lifestyle characteristics at baseline, including age, sex, alcohol consumption, smoking status, physical activity, monthly income, education level, marital status, and menopausal status. Anthropometric measurements were conducted by trained research staff. Body mass index (BMI) was calculated as body weight (kg) divided by the square of height (m^2^).

### 2.7. Statistical Analysis

The descriptive statistics are presented as the mean ± standard error for continuous variables and as numbers (percentages) for categorical variables. The generalized linear model was used to compare the differences in the means of baseline characteristics and to test for the linear trends across the quintiles of seafood consumption. A Chi-square test or Fisher’s exact test was used to determine the differences in the distributions of general characteristics of the study participants. Spearman correlations were used to assess the relationship between the CDAI and BDI scores. Because CDAI scores are not observed as normally distributed but rather with skewed distribution, they were transformed by using the natural log before analysis. Associations between seafood consumption and depression were estimated from hazard ratios (HRs) and 95% confidence intervals (CIs) by using Cox’s proportional hazard model. Person-years were calculated from the date they completed the baseline examination to the date of depression onset or the end of follow-up. For adjustment in the multivariable model, potential confounders from the previously published scientific literature were taken into account [9,29,30,31,32] with stepwise regression procedures, including age (continuous), BMI (continuous), alcohol consumption (nondrinker, former drinker, and current drinker), smoking status (nonsmoker, former smoker, and current smoker), physical activity(<30 min/day/≥30 min/day), monthly income (<200 million Korean won (KRW)/≥200 million KRW), education level (<college/≥college), marital status (married/other), menopausal status (premenopause/menopause, current hormone replacement therapy (HRT) use/menopause, past HRT use/menopause, non-HRT use/menopause, unknown HRT use), consumption of fruit, vegetables, and meat (quintile), and total energy intake (continuous). We additionally adjusted for the baseline BDI score (continuous) for men. All analyses were performed using SAS version 9.4 (SAS Institute Inc., Cary, NC, USA). Statistical significance was considered at *p* < 0.05. Data were stratified according to sex, as previous research reported that sex influences the association between seafood consumption and depression [6,29].

## 3. Results

### 3.1. Baseline Characteristics

During a follow-up time of 8 years, 165 (11.9%) men and 224 (18.9%) women experienced depression. The baseline characteristics of participants according to quintiles of seafood consumption are described in Table 1. Among men, participants with higher seafood consumption had a lower baseline BDI score, had a higher mean BMI, had a higher proportion of current drinkers, were more physically active, and had higher levels of education (*p* < 0.05). In contrast, women showed no significant difference in BDI score among the groups. Among women, participants with higher seafood consumption were more likely to be younger, have a higher mean BMI, be more physically active, and have higher levels of monthly income and education (*p* < 0.05).

### 3.2. Dietary Intakes

Table 2 and Table 3 present food consumption (g/1000 kcal) and nutrient intake per 1000 kcal according to quintiles of seafood consumption. There was a positive association between seafood consumption and most of the food groups (all *p* < 0.05) in both men and women. However, as seafood consumption increased, the consumption of grains (*p* < 0.001) decreased in both men and women. In addition, as seafood consumption increased, the consumption of oils and sugars (*p* < 0.05) decreased among women.

Regarding nutrient intake, participants with higher seafood consumption showed a higher total energy intake (*p* < 0.001) in both men and women. In line with the above results, participants with higher seafood consumption showed higher intakes of most nutrients (all *p* < 0.05) for both sexes. In contrast, there was a negative association between seafood consumption and carbohydrate intake (*p* < 0.001) in both men and women.

### 3.3. Associations Between Seafood Consumption and the Risk of Depression

HRs with 95% CIs for the associations between seafood consumption and depression distributed by quintiles are presented in Table 4. After adjustment for potential confounders, the risk of depression was inversely associated with seafood consumption, with a 42% lower risk (HR _T5 vs. T1_ = 0.58, 95% CI: 0.35–0.98, *p* = 0.040) in the highest quintile of seafood consumption compared with the reference group among women. Seafood consumption was not significantly associated with the risk of depression among men.

### 3.4. Correlations Between CDAI and BDI Scores According to N-3 PUFA Status

The correlation between CDAI and BDI scores is shown in Table 5. In the group with a high *n*-3 PUFA intake, a significant inverse correlation was found between CDAI and BDI scores (r = −0.146, *p* < 0.001) among women. There was no correlation between CDAI and BDI scores among men.

## 4. Discussion

In this prospective cohort study of Korean middle-aged men and women, we examined seafood consumption in relation to risk of depression. Seafood consumption, which comprised fish, shellfish, and seaweed was inversely associated with risk of depression among women when comparing the highest and lowest quintiles after multivariable adjustments. Moreover, in the group with high *n*-3 PUFA intake, CDAI scores were negatively correlated with BDI scores among women.

In our study, socioeconomic status such as higher levels of education and income was associated with greater consumption of seafood. A systematic review of European countries revealed that the socioeconomic factors might influence dietary habits [33]. A study analyzing data of 5721 participants from the Korea National Health and Nutrition Examination Survey, reported that higher socioeconomic status was associated with the “modified” dietary pattern, which reflected good nutritional status [34]. A cross-sectional study delineating yearly trends in the daily consumption of seafood and investigating the socioeconomic factors influencing seafood consumption among elderly Koreans, showed that there was a significant correlation between seafood intake and educational level and family income [35]. Considering the possibility that socioeconomic factors might result in different dietary habits, thus affecting the risk of depression, we adjusted for these potential confounders.

Our findings showed that the risk of depression was inversely associated with seafood consumption among women. This association remained significant after adjusting for the effects of fruit and vegetable intakes, which are considered as important confounding factors. A cross-sectional study in Korea demonstrated that the highest tertile of seafood consumption was associated with a decreased risk of depression compared to the lowest tertile [30] in men and women. A population-based cohort study of older adults in Tuscany (Italy) indicated that a high intake of fish and shellfish was prospectively associated with a decrease in depressive symptoms 3 years later [31]. Subjects who ate fish ≥2 times/week at baseline had a 25% lower risk of depression during follow-up than those who ate fish <2 times/week in the longitudinal study of Australian adults among women [32]. A meta-analysis revealed that an increment of 1 serving/week of fish consumption was associated with 11% lower risk of depression [6].

Particularly, EPA and DHA, which are the most abundant *n*-3 PUFA present in seafood, might be the main drivers of the associations between seafood intake and the risk of depression. A meta-analysis of observational studies showed that both total *n*-3 PUFA and fish-derived *n*-3 PUFA were associated with decreased risk of depression [12]. Higher intakes of total *n*-3 PUFA, DHA, and EPA were associated with lower odds of depressive symptoms in the Supplementation with Antioxidant Vitamins and Minerals Study conducted in France [36]. A cohort study conducted in Japan reported that an increased intake of EPA and DHA was inversely associated with the risk of depressive symptoms, as well as *n*-3 PUFA [37]. Total fish consumption, EPA and DHA had a reverse J-shaped association with the risk of psychiatrist-diagnosed major depressive disorder in Japanese cohort study [38].

The beneficial effects of seafood and its components on depression might be explained by numerous mechanisms. The imbalance of neurotransmission plays an important role in the pathophysiology of depression. Intake of *n*-3 PUFAs positively influences the depressive status by maintaining the membrane structure and functions of brain, which may potentially modulate the serotoninergic and dopaminergic transmission [13]. The highly unsaturated nature of EPA and DHA affects membrane fluidity of several types of cells [39,40]. *N*-3 PUFAs also regulate the signal transductions by inducing membrane changing, such as stimulating the activity of diacylglycerol kinase [41] and Na/K-dependent ATPase [42]. Beside these neurotransmitter system as the underlying mechanisms of major depression, alterations in glutamatergic system have been implicated in age-related cognitive deficits [43,44]. The N-methyl-D-aspartate receptor, a glutamate receptor, is a binding or modulation site for antidepressant [45]. In experimental animals, deficiency of *n*-3 PUFAs aggravates the reductions in glutamatergic synaptic efficacy and its astroglial regulation in the hippocampal CA1, which is involved in spatial memory [43].

In the present study, as seafood consumption increased, the consumption of grains, oils and sugar decreased among women. Moreover, there was a negative association between seafood consumption and carbohydrate intake in both men and women. A study utilizing data of 75,466 participants from UK Biobank reported that the dietary pattern, which is characterized by high intakes of grain-based desserts, chocolate and confectionery, and butter, is associated with a higher risk of depressive and anxious symptoms [46]. The observational retrospective study conducted in Spain showed that a high consumption of sweet foods and refined sugars was significantly associated with depression [47]. Some studies have demonstrated that low carbohydrate intakes are correlated with a decreased risk of depression [48,49]. A cross-sectional study of United States adults reported that the low-carbohydrate-diet score, which provide a comprehensive assessment of diets with a lower intake of carbohydrate and a higher intake of protein and fat, was inversely associated with the risk of depression [50].

Dysregulation of the functional activity of the peripheral immune system is observed in major depression, which is characterized by increased levels of proinflammatory cytokines [13]. Eicosanoids produced from *n*-3 PUFAs affect inflammation and regulation of immune function through incorporation in cell membrane, which results in the release of 20 carbon arachidonic (AA) content from membrane phospholipids. This procedure subsequently leads to the reduction on the amount of substrate available to produce inflammatory and immunoregulatory eicosanoids [51]. Beside their action on eicosanoids, *n*-3 PUFA have been reported to decrease proinflammatory cytokines production, such as TNF-α and interleukin-6 [52]. Increased intake of refined carbohydrates and refined vegetable oils rich in omega-6 fatty acids led to the production of AA, thereby leading to elevated inflammation in various organs [53]. Dietary sugars may also elicit inflammatory processes. Fructose-fed rats had increased visceral adipose tissue mass along with elevated levels of inflammatory factors and increased expression of inflammatory genes [54]. Chronic stress, which might induce depressive-like behaviors, exacerbated blood–brain barrier damage and increased neuroinflammation in high fructose diet-fed mice [55].

Our study showed that in the group with high *n*-3 PUFA intake, significant inverse correlation was found between CDAI and BDI scores among women. As discussed above, *n*-3 PUFAs can exert the potential anti-inflammatory effects via preventing or decreasing the inflammatory status. Also, numerous studies have demonstrated that a protective effect of consuming a diet rich in antioxidants on the risk of depression [56,57]. In experimental animals, combined treatment of *n*-3 PUFA and ascorbic acid provided an additive effect in suppressing lipid peroxidation compared to *n*-3 PUFA or ascorbic acid alone [58]. A recent meta-analysis of human, which pooled data from 10 randomized controlled trials of *n*-3 fatty acids in patients with acute inflammatory lung injury, suggested that the enteral formulation which provided *n*-3 fatty acids in combination with antioxidant and γ-linolenic acid formulation led to more favorable outcomes [17]. Regarding the results, it might be suggested that *n*-3 PUFA could exert more beneficial effects against depression, which is associated with inflammatory conditions, if accompanied by increased consumption of foods rich in antioxidants.

Sex differences were also observed in our study. Generally, women were more likely to engage in health-promoting behaviors and have healthier lifestyle patterns than men [59]. According to our results, the consumption of plant-based foods, such as fruits and vegetables, was higher among women than among men. A previous intervention study designed to investigate sex hormone effects on *n*-3 highly unsaturated fatty acids in human subjects, revealed that DHA concentrations in plasma cholesteryl esters were higher in women than in men and that this difference was independent of dietary differences. Estrogen induced an increase in DHA status in women, probably by regulating biosynthesis of DHA [60]. These sex-dependent variations might affect the reduced risk of depression in women compared to men.

The main strength of the current study was the design as a prospective long-term follow-up for up to 8 years. Moreover, we had comprehensive information on potential confounding factors based on a questionnaire administered by skilled interviewers. Nonetheless, the study has some limitations. First, we had no data regarding to depression treatment, such as antidepressants and medication compliance. Second, self-reported dietary data might have some information bias. Third, we assessed dietary intakes only at baseline did not capture changes in the diet over the course of the follow-up period. Finally, although the study setting accounted for potential confounders, the generalizability of our results may be limited. Therefore, our results need to be replicated in other population groups with repeated dietary assessments and more diverse ethnic backgrounds.

## 5. Conclusions

In conclusion, findings from this longitudinal study suggest that seafood consumption was inversely associated with risk of depression in Korean women. Moreover, *n*-3 PUFA in seafood might have a protective effect against depression if accompanied by increased consumption of foods rich in antioxidants. Our findings will provide an important basis to further examine the benefits of seafood consumption for mental health.

## Figures and Tables

**Figure 1 antioxidants-13-01413-f001:**
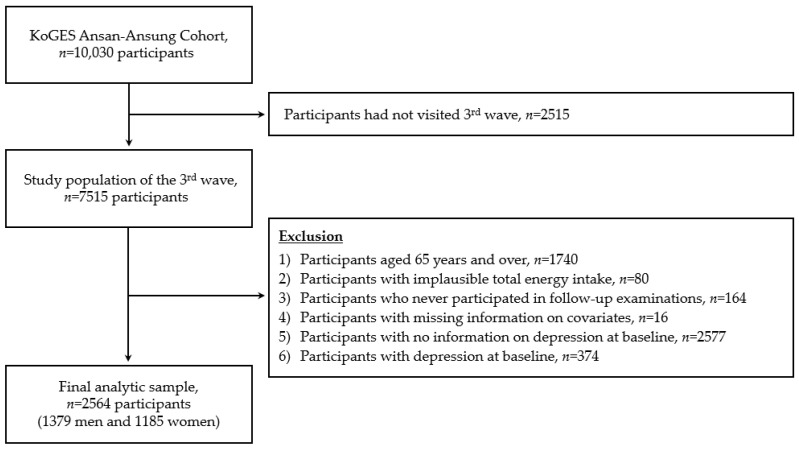
A flowchart of the study population.

**Table 1 antioxidants-13-01413-t001:** General characteristics according to seafood-consumption quintile.

	Quintile 1		Quintile 2		Quintile 3		Quintile 4		Quintile 5		*p*-Value ^†^
Men											
No. of participants	275		276		276		276		276		
Median (g/d)	14.8		26.1		37.3		52.2		81.4		
Range (g/d)	0.4–20.7		20.8–31.3		31.3–44.8		44.9–62.4		62.4–272.9		
BDI score	5.3 ± 0.2		5.5 ± 0.2		5.9 ± 0.3		5.7 ± 0.3		4.9 ± 0.2		0.035
Age (years)	51.3 ± 0.3		50.6 ± 0.3		50.6 ± 0.3		50.9 ± 0.3		50.3 ± 0.3		0.272
BMI (kg/m^2^)	24.1 ± 0.2		24.7 ± 0.2		24.3 ± 0.2		24.8 ± 0.2		25.3 ± 0.2		<0.001
Current drinker	182 (66.2)		206 (74.6)		208 (75.4)		213 (77.2)		215 (77.9)		0.012
Current smoker	77 (28.0)		81 (29.4)		81 (29.4)		93 (33.7)		102 (37.0)		0.127
Physical activity, ≥30 min/d	136 (49.5)		164 (59.4)		149 (54.0)		156 (56.5)		171 (62.0)		0.032
Monthly income, ≥2 million KRW	206 (74.9)		219 (79.4)		224 (81.2)		223 (80.8)		232 (84.1)		0.102
Education, ≥college	83 (30.2)		82 (29.7)		104 (37.7)		104 (37.7)		119 (43.1)		0.003
Marital status, married	265 (96.4)		266 (96.4)		272 (98.6)		272 (98.6)		269 (97.5)		0.257
Women											
No. of participants	237		237		237		237		237		
Median (g/d)	12.6		21.5		32.6		44.9		72.2		
Range (g/d)	0.3–17.0		17.1–26.9		26.9–37.7		37.8–55.7		55.7–224.9		
BDI score	7.1 ± 0.3		6.5 ± 0.3		6.0 ± 0.3		6.3 ± 0.3		6.2 ± 0.3		0.067
Age (years)	51.3 ± 0.4		50.5 ± 0.4		50.7 ± 0.4		49.9 ± 0.3		50.1 ± 0.3		0.033
BMI (kg/m^2^)	24.7 ±0.2		23.9 ± 0.2		24.5 ± 0.2		24.3 ± 0.2		24.8 ± 0.2		0.008
Current drinker	60 (25.3)		67 (28.3)		78 (32.9)		65 (27.4)		75 (31.7)		0.347
Current smoker	4 (1.7)		3 (1.3)		2 (0.8)		5 (2.1)		5 (2.1)		0.809
Physical activity, ≥30 min/d	114 (48.1)		131 (55.3)		136 (57.4)		132 (55.7)		153 (64.6)		0.010
Monthly income, ≥2 million KRW	142 (59.9)		153 (64.6)		169 (71.3)		183 (77.2)		174 (73.4)		<0.001
Education, ≥college	23 (9.7)		38 (16.0)		42 (17.7)		27 (11.4)		51 (21.5)		0.002
Marital status, married	210 (88.6)		214 (90.3)		220 (92.8)		226 (95.4)		217 (91.6)		0.085
Menopausal status											0.296
Premenopause	98 (41.4)		126 (53.2)		111 (46.8)		120 (50.6)		131 (55.3)		
Menopause, current HRT use	11 (4.6)		11 (4.6)		13 (5.5)		12 (5.1)		11 (4.6)		
Menopause, past HRT use	28 (11.8)		19 (8.0)		29 (12.2)		32 (13.5)		28 (11.8)		
Menopause, non-HRT use	89 (37.6)		75 (31.7)		77 (32.5)		66 (27.9)		62 (26.2)		
Menopause, unknown HRT use	11 (4.6)		6 (2.5)		7 (3.0)		7 (3.0)		5 (2.1)		

BDI, Beck Depression Inventory; BMI, body mass index; KRW, Korean won; HRT, hormone replacement therapy. The values are expressed as the mean ± SE for continuous variables and numbers (percentages) for categorical variables. ^†^ The *p*-value was obtained from the general linear models for continuous variables and a Chi-square or Fisher’s exact test for categorical variables.

**Table 2 antioxidants-13-01413-t002:** Food consumption according to seafood-consumption quintile.

Food Consumption (g/1000 kcal)	Quintile 1		Quintile 2		Quintile 3		Quintile 4		Quintile 5		*p*-Trend ^†^
	Mean	SE	Mean	SE	Mean	SE	Mean	SE	Mean	SE	
Men											
Grains	438.9	3.7	411.6	3.7	410.0	4.0	383.7	3.4	359.4	4.1	<0.001
Potatoes	5.6	0.4	6.0	0.4	6.8	0.5	6.6	0.5	7.0	0.5	0.019
Legumes	19.7	1.5	20.8	1.5	21.2	1.5	21.1	1.2	23.7	1.4	0.052
Nuts and seeds	0.4	0.1	0.5	0.1	0.6	0.1	0.8	0.1	0.6	0.1	0.010
Fruits	104.0	4.0	98.5	3.9	117.6	4.9	117.8	4.5	133.5	5.5	<0.001
Vegetables	30.4	1.1	33.5	1.1	36.1	1.0	41.9	1.3	45.6	1.5	<0.001
Mushrooms	3.6	0.2	3.4	0.2	3.7	0.2	5.0	0.3	5.3	0.4	<0.001
Meats	19.8	0.8	25.0	1.0	27.6	1.0	30.9	1.1	31.9	1.2	<0.001
Eggs	6.0	0.4	6.6	0.4	6.6	0.4	6.8	0.5	7.5	0.5	0.019
Fish and shellfish	7.9	0.2	13.6	0.2	18.6	0.3	25.5	0.4	40.4	1.0	<0.001
Seaweeds	0.5	0.0	0.6	0.0	0.6	0.0	0.7	0.0	0.8	0.1	<0.001
Milk and dairy products	54.7	3.8	53.5	3.3	50.5	3.2	53.0	3.3	60.7	3.6	0.152
Oils and sugars	3.8	0.2	4.2	0.2	3.3	0.2	3.7	0.2	3.6	0.2	0.206
Women											
Grains	429.0	4.9	406.3	4.7	384.9	4.7	369.3	4.2	342.0	5.0	<0.001
Potatoes	9.9	0.7	10.0	0.6	10.8	0.7	10.8	0.7	11.7	0.7	0.037
Legumes	18.8	1.5	20.8	1.7	25.0	1.6	22.8	1.4	24.1	1.3	0.017
Nuts and seeds	0.7	0.2	0.7	0.1	0.6	0.1	0.9	0.1	1.0	0.1	0.010
Fruits	159.6	6.7	161.4	5.9	196.3	8.0	179.9	6.3	192.1	6.9	<0.001
Vegetables	38.7	1.6	43.0	1.6	47.6	1.5	50.7	1.6	62.4	2.2	<0.001
Mushrooms	4.2	0.3	5.0	0.3	4.7	0.4	5.8	0.3	7.1	0.4	<0.001
Meats	14.6	0.9	17.9	0.9	18.1	0.9	19.6	0.9	22.3	1.0	<0.001
Eggs	6.3	0.4	6.5	0.4	7.1	0.5	7.6	0.5	7.1	0.4	0.097
Fish and shellfish	7.6	0.2	13.4	0.3	18.7	0.3	25.3	0.5	41.1	1.2	<0.001
Seaweeds	0.7	0.0	0.9	0.1	0.9	0.1	1.0	0.1	1.2	0.1	<0.001
Milk and dairy products	73.5	5.0	75.1	4.6	88.2	5.4	80.0	4.3	88.7	4.5	0.022
Oils and sugars	2.5	0.2	2.2	0.2	2.1	0.2	2.2	0.2	2.0	0.2	0.048

The values are expressed as the mean ± SE. ^†^ The *p*-trend was calculated by treating the median value of each quintile as a continuous variable in the general linear model.

**Table 3 antioxidants-13-01413-t003:** Nutrient intake according to seafood-consumption quintile.

	Quintile 1		Quintile 2		Quintile 3		Quintile 4		Quintile 5		*p*-Trend ^†^
	Mean	SE	Mean	SE	Mean	SE	Mean	SE	Mean	SE	
Men											
Total energy (kcal/d)	1722	22.8	1901	22.7	2041	26.2	2108	27.7	2339	34.2	<0.001
Protein (g/d)	29.7	0.2	31.4	0.2	32.6	0.2	34.5	0.3	37.6	0.3	<0.001
Fat (g/d)	14.2	0.3	16.0	0.3	16.5	0.3	17.6	0.3	18.7	0.3	<0.001
Carbohydrate (g/d)	185.2	0.7	179.9	0.7	177.5	0.7	173.3	0.8	168.1	0.8	<0.001
Vitamin A (RE/d)	201.6	5.4	220.7	5.4	227.1	5.4	243.4	5.5	266.6	7.0	<0.001
Vitamin B_1_ (mg/d)	0.5	0.0	0.6	0.0	0.6	0.0	0.6	0.0	0.6	0.0	<0.001
Vitamin B_2_ (mg/d)	0.5	0.0	0.5	0.0	0.5	0.0	0.5	0.0	0.6	0.0	<0.001
Niacin (mg/d)	7.3	0.1	7.8	0.1	7.9	0.1	8.6	0.1	9.2	0.1	<0.001
Vitamin C (mg/d)	49.9	1.3	51.3	1.4	52.9	1.2	57.3	1.3	61.8	1.4	<0.001
Zinc (µg/d)	3.9	0.0	4.2	0.0	4.3	0.0	4.5	0.0	4.9	0.0	<0.001
Vitamin B_6_ (mg/d)	0.8	0.0	0.8	0.0	0.8	0.0	0.9	0.0	0.9	0.0	<0.001
Folate (µg/d)	105.4	1.9	108.0	2.0	108.5	1.9	117.2	2.1	123.5	2.2	<0.001
Retinol (µg/d)	27.2	1.3	30.7	1.1	31.6	1.0	35.8	1.2	44.2	1.4	<0.001
Carotene (µg/d)	1012	29.3	1100	30.9	1128	30.8	1203	31.5	1283	40.3	<0.001
Fiber (g/d)	3.1	0.1	3.0	0.1	3.1	0.1	3.1	0.1	3.2	0.1	0.013
Vitamin E (mg/d)	3.9	0.1	4.2	0.1	4.4	0.1	4.8	0.1	5.2	0.1	<0.001
Women											
Total energy (kcal/d)	1519	25.5	1628	26.3	1735	25.7	1843	28.2	2040	33.3	<0.001
Protein (g/d)	29.7	0.3	31.0	0.3	32.8	0.3	34.2	0.3	37.8	0.4	<0.001
Fat (g/d)	13.0	0.3	14.1	0.3	15.0	0.3	16.1	0.3	17.7	0.3	<0.001
Carbohydrate (g/d)	188.5	0.8	184.8	0.8	181.9	0.8	178.1	0.8	171.6	0.9	<0.001
Vitamin A (RE/d)	219.9	7.3	229.1	7.8	257.2	7.7	267.7	7.7	313.4	9.9	<0.001
Vitamin B_1_ (mg/d)	0.5	0.0	0.5	0.0	0.6	0.0	0.6	0.0	0.6	0.0	<0.001
Vitamin B_2_ (mg/d)	0.5	0.0	0.5	0.0	0.5	0.0	0.6	0.0	0.6	0.0	<0.001
Niacin (mg/d)	7.1	0.1	7.4	0.1	7.7	0.1	8.1	0.1	8.9	0.1	<0.001
Vitamin C (mg/d)	63.0	1.8	63.8	1.6	73.5	2.1	73.2	2.0	81.3	2.2	<0.001
Zinc (µg/d)	4.0	0.0	4.2	0.0	4.3	0.0	4.4	0.1	4.7	0.1	<0.001
Vitamin B_6_ (mg/d)	0.8	0.0	0.9	0.0	0.9	0.0	0.9	0.0	1.0	0.0	<0.001
Folate (µg/d)	116.8	2.6	116.7	2.5	126.8	2.8	130.6	2.6	143.2	2.8	<0.001
Retinol (µg/d)	29.6	1.3	33.3	1.4	37.8	1.5	40.5	1.4	44.7	1.5	<0.001
Carotene (µg/d)	1120	43.0	1145	44.1	1286	44.0	1325	43.3	1569	57.9	<0.001
Fiber (g/d)	3.4	0.1	3.3	0.1	3.6	0.1	3.6	0.1	3.8	0.1	<0.001
Vitamin E (mg/d)	4.2	0.1	4.4	0.1	4.9	0.1	5.2	0.1	5.9	0.1	<0.001

RE, retinol equivalents. The values are expressed as the mean ± SE. Nutrient intakes were expressed per 1000 kcal. ^†^ The *p*-trend was calculated by treating the median value of each quintile as a continuous variable in the general linear model.

**Table 4 antioxidants-13-01413-t004:** Hazard ratios and 95% confidence intervals for depression according to the seafood consumption quintile.

					Model 1 *			Model 2 ^†^	
	No. of Cases (%)		Person-Years	HR	95% CI	*p*-Value	HR	95% CI	*p*-Value
Men									
Quintile 1 (*n* = 275)	32 (11.6)		1907.2	1.00 (Reference)		-	1.00 (Reference)		-
Quintile 2 (*n* = 276)	26 (9.4)		1833.9	0.85	0.51–1.43	0.543	0.85	0.50–1.44	0.534
Quintile 3 (*n* = 276)	37 (13.4)		1930.9	1.15	0.72–1.84	0.569	1.00	0.60–1.67	0.988
Quintile 4 (*n* = 276)	31 (11.2)		1889.4	0.98	0.60–1.61	0.938	0.89	0.52–1.54	0.675
Quintile 5 (*n* = 276)	39 (14.1)		1895.5	1.23	0.77–1.97	0.380	1.28	0.73–2.24	0.386
Women									
Quintile 1 (*n* = 237)	54 (22.8)		1543.7	1.00 (Reference)		-	1.00 (Reference)		-
Quintile 2 (*n* = 237)	45 (19.0)		1601.9	0.78	0.53–1.16	0.220	0.83	0.55–1.25	0.377
Quintile 3 (*n* = 237)	51 (21.5)		1606.6	0.89	0.60–1.30	0.532	0.94	0.62–1.42	0.775
Quintile 4 (*n* = 237)	45 (19.0)		1667.8	0.75	0.50–1.11	0.149	0.87	0.56–1.34	0.517
Quintile 5 (*n* = 237)	31 (13.1)		1669.3	0.49	0.32–0.77	0.001	0.58	0.35–0.98	0.040

HR, hazard ratio; CI, confidence interval. * Model 1 was unadjusted. ^†^ Model 2 was adjusted for energy intake, age, BMI, baseline BDI score, current drinker, current smoker, physical activity, monthly income, education, marital status, and consumption of fruit, vegetables, and meat (men). Model 2 was adjusted for energy intake, age, BMI, current drinker, current smoker, physical activity, monthly income, education, marital status, and consumption of fruit, vegetables, and meat (women).

**Table 5 antioxidants-13-01413-t005:** Correlation between composite dietary antioxidant index and Beck Depression Inventory scores according to *n*-3 PUFAs status.

	Men			Women	
	r	*p*		r	*p*
*n*-3 PUFAs (g/d)					
Low (<1.14) *	−0.021	0.611	Low (<1.07)	−0.030	0.492
High (≥1.14)	−0.051	0.217	High (≥1.07)	−0.146	<0.001

PUFAs, polyunsaturated fatty acids. * The participants were divided into each group, with low and high being based on a median value.

## Data Availability

Data in this study were available from the Korean Genome and Epidemiology Study (KoGES; 6635-302) conducted by the National Institute of Health, Centers for Disease Control and Prevention, Ministry for Health and Welfare, Republic of Korea.

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
