# Peer review of "N-3 Fatty Acids in Seafood Influence the Association Between the Composite Dietary Antioxidant Index and Depression: A Community-Based Prospective Cohort Study"

_antioxidants, 2024, doi:10.3390/antiox13111413_

Round 1

Reviewer 1 Report

Thank you for this nice manuscript which is well written and provides intresting data. 

I am not fully convinced of the detailed interpretation of the data, since data collection might have been biased (FFQ). However, the study contributes with more information to the topic of n3- fatty acids and depression. 

Thank you for this nice manuscript which is well written and provides intresting data. 

I have some minor remarks:

Methods: 

- Please avoid the use of active and passive voice- decide for one. 

- Why did you exclude people older than 65 years? 

- BDI cut-off of 16 seems too low to me, but might be true for the Korean population; however, no sex difference was made here. Normally, the cut-off is higher in women (Schmitt et al., Diagnostica, 2006)

- Assessment of food consumption is always a challenge. Using FFQs is common but contains sources of errors: FFQ over one past year seems too vague to calculate caloric intake and even nutrient intake. Further, it can be seen that the intake of nearly all food groups is higher in the highest quitile group: maybe there is a systematic bias overestimating the intake of specific foods. 

 - Which software did you use for calculating nutrients/ calories etc.?

- The classification of alcohol consumption is not sensible to me. Why didn't you ask for quantity? 

- How much are 200 million KRW in USDollar or Euro? 

Results:

Tab. 1: please clarify median and range ("of sea food consumption"?!). 

KRW should be in the line above

 Tab.5 should be arranged more clearly

Discussion:

- typo in line 15 (<2 times/ week?)

- typo in line 55 (depressive and anxiety?)

- revise sentence in line 79-80

l. 74-95: whole paragraph is not clear to me. It is beyond the topic of the paper (depression) and seems for me redundant here.

Author Response

Manuscript Number: antioxidants-3230235

Title: N-3 fatty acids in seafood influence the association between composite dietary antioxidant index and depression: a community-based prospective cohort study

Responses to Reviewer's comments (Reviewer 1)

We thank the reviewer for careful reading and description about our manuscript with the valuable comments. We worked to the best of our abilities to revise the issues reviewer point out.

Major comments: Thank you for this nice manuscript which is well written and provides interesting data. I am not fully convinced of the detailed interpretation of the data, since data collection might have been biased (FFQ). However, the study contributes with more information to the topic of n3- fatty acids and depression.

Response: The SQFFQ used in the study is a validated tool designed to assess the frequency and amount of commonly consumed food items among middle-aged Koreans over the past year. The reproducibility and validity was tested using 3-day dietary records as a reference method during each of the four seasons over a period of 1 year, totaling 12 days for 124 participants (Ahn Y, Kwon E, Shim JE, et al. Validation and reproducibility of food frequency questionnaire for Korean genome epidemiologic study. Eur J Clin Nutr. 2007;61(12):1435-1441. doi:10.1038/sj.ejcn.1602657). Therefore, we decided that the SQFFQ could be an acceptable tool for assessing usual food intake in the Korean population.

Detail comments: Thank you for this nice manuscript which is well written and provides interesting data. I have some minor remarks:

Methods:

- Please avoid the use of active and passive voice- decide for one.

Response: : As the reviewer stated, we rephrased overall text consistently.

- Why did you exclude people older than 65 years?

Response: Definitions of the older population vary across the world because of differences in age distribution. Generally, “older population” refers to those aged 65 and over and “working-age population” refers to ages 20 to 64 (HE, Wan. An aging world: 2015. US Department of Commerce․ Economics and Statistics Administration, 2016.). Therefore, we excluded participants aged 65 and over for the purpose of focusing on middle-aged Koreans.

- line 83-84

Among the 7,515 participants, we excluded participants aged 65 and over for the purpose of focusing on middle-aged Koreans (n=1,740).

- BDI cut-off of 16 seems too low to me, but might be true for the Korean population; however, no sex difference was made here. Normally, the cut-off is higher in women (Schmitt et al., Diagnostica, 2006)

Response: We truly appreciate with the reviewer's suggestion. The validity and reliability of the BDI in the Korean population was previously established and suggested a cut-off point of 16 for depression. Also, cut-off score of the evaluation criteria are often not different according to sex for Koreans (Kim SG, Park J, Kim HT, Pan Z, Lee Y, McIntyre RS. The relationship between smartphone addiction and symptoms of depression, anxiety, and attention-deficit/hyperactivity in South Korean adolescents. Ann Gen Psychiatry. 2019;18:1. Published 2019 Mar 9. doi:10.1186/s12991-019-0224-8; Park SJ, Kim MS, Lee HJ. The association between dietary pattern and depression in middle-aged Korean adults. Nutr Res Pract. 2019;13(4):316-322. doi:10.4162/nrp.2019.13.4.316).

- Assessment of food consumption is always a challenge. Using FFQs is common but contains sources of errors: FFQ over one past year seems too vague to calculate caloric intake and even nutrient intake. Further, it can be seen that the intake of nearly all food groups is higher in the highest quitile group: maybe there is a systematic bias overestimating the intake of specific foods.

Response: The reproducibility and validity of FFQ was tested using 3-day dietary records as a reference method during each of the four seasons over a period of 1 year, totaling 12 days for 124 participants. In conclusion, the FFQ appeared to be accurate, thus to be an acceptable tool for assessing the nutrient intakes. (Ahn Y, Kwon E, Shim JE, et al. Validation and reproducibility of food frequency questionnaire for Korean genome epidemiologic study. Eur J Clin Nutr. 2007;61(12):1435-1441. doi:10.1038/sj.ejcn.1602657). Also, all nutrient and food intakes were adjusted for total energy intake by density methods(unit/1,000 kcal) to avoid overestimating the intake of specific foods in our study.

- Which software did you use for calculating nutrients/ calories etc.?

Response: Daily nutrient intakes and calories were calculated from the food intake measured by SQFFQ using the computer-aided nutritional analysis program (CAN Pro), a nutrient database developed by the Korean Nutrition Society.

- line 123-125

Daily nutrient intakes and calories were calculated from the food intake measured by SQFFQ using the computer-aided nutritional analysis program (CAN Pro), a nutrient database developed by the Korean Nutrition Society (27).

- The classification of alcohol consumption is not sensible to me. Why didn't you ask for quantity?

Response: We agreed that it might be more appropriate if alcohol consumption was classified based on quantity. In cohort used in our study, alcohol consumption data were collected by trained interviewers using alcoholic beverage questionnaire. Participants were asked to report their drinking status (current-, past-, or nondrinkers). Only for current drinkers, drinking habit such as average frequency of drinking during the month for six liquors (raw rice wine, beer, refined rice wine, wine, soju, and whisky) and the usual amount when they drank, were examined. Therefore, for alcohol consumption, we classified the participants depending on their drinking status, not the quantity. Furthermore, other studies using the same cohort design classified the alcohol consumption according to status (Chae MJ, Jang JY, Park K. Association between dietary calcium intake and the risk of cardiovascular disease among Korean adults. Eur J Clin Nutr. 2020;74(5):834-841. doi:10.1038/s41430-019-0525-7/Kong SH, Kim JH, Hong AR, Cho NH, Shin CS. Dietary calcium intake and risk of cardiovascular disease, stroke, and fracture in a population with low calcium intake. Am J Clin Nutr. 2017;106(1):27-34. doi:10.3945/ajcn.116.148171).

- How much are 200 million KRW in USDollar or Euro?

Response: Considering the current exchange rate, 200 million KRW equals about 1430.41 US dollars and 1335.11 Euro.

Results:

- Tab.1: please clarify median and range ("of sea food consumption"?!).

Response: As the reviewer stated, we clarified the Table 1.

â”” KRW should be in the line above

Response: As the reviewer stated, we revised that part.

- Tab.5 should be arranged more clearly

Response: As the reviewer stated, we arranged the Table 5 more clearly.

Discussion:

- typo in line 15 (<2 times/ week?)

Response: As the reviewer stated, we revised that part.

- line 28-30

Subjects who ate fish ≥2 times/week at baseline had a 25% lower risk of depression during follow-up than those who ate fish <2 times/week in the longitudinal study of Australian adults among women (32). 

- typo in line 55 (depressive and anxiety?)

Response: As the reviewer stated, we revised that part.

- line 60-64

A study utilizing data of 75,466 participants from UK Biobank reported that the dietary pattern, which is characterized by high intakes of grain-based desserts, chocolate and confectionery, and butter, is associated with a higher risk of depressive and anxious symptoms (46).

- revise sentence in line 79-80

Response: As the reviewer stated, we revised that part.

- line 89-90

Also, numerous studies have demonstrated that a protective effect of consuming a diet rich in antioxidants on the risk of depression (56, 57).

 â””74-95: whole paragraph is not clear to me. It is beyond the topic of the paper (depression) and  seems for me redundant here.

Response: As the reviewer stated, we rearranged that part and focused on the main findings.

- line 86-99

Our study showed that in the group with high n-3 PUFA intake, significant inverse correlation was found between CDAI and BDI scores among women. As discussed above, n-3 PUFAs can exert the potential anti-inflammatory effects via preventing or decreasing the inflammatory status. Also, numerous studies have demonstrated that a protective effect of consuming a diet rich in antioxidants on the risk of depression (56, 57). In experimental animals, combined treatment of n-3 PUFA and ascorbic acid provided an additive effect in suppressing lipid peroxidation compared to n-3 PUFA or ascorbic acid alone (58). A recent meta-analysis of human, which pooled data from 10 randomized controlled trials of n-3 fatty acids in patients with acute inflammatory lung injury, suggested that the enteral formulation which provided n-3 fatty acids in combination with antioxidant and γ-linolenic acid formulation led to more favorable outcomes (17). Regarding the results, it might be suggested that n-3 PUFA could exert more beneficial effects against depression, which is associated with inflammatory conditions, if accompanied by increased consumption of foods rich in antioxidants.

Reviewer 2 Report

The authors examined longitudinal associations between seafood consumption, PUFA, EPA, DHA and depression. While the research contributes to understanding these associations over time, I do not believe that research in this area is original because multiple results have been achieved to date. However, this study does address a specific gap in the field. The strength of this study is the point in time when the long-term course was followed. The impact of the long-term course of the study was evaluated in comparison with the results of previous studies. As for methodology, I think the current cohort is unlikely to improve any further. The conclusions are consistent with the evidence and arguments presented. The references used are appropriate. I only have one concern, please see the detail comments.

The results of this study suggest that economic status, such as job and annual income, may affect depression. The possibility that a higher annual income may result in different dietary habits should be considered.

Author Response

Manuscript Number: antioxidants-3230235

Title: N-3 fatty acids in seafood influence the association between composite dietary antioxidant index and depression: a community-based prospective cohort study

Responses to Reviewer's comments (Reviewer 2)

Major comments: The authors examined longitudinal associations between seafood consumption, PUFA, EPA, DHA and depression. While the research contributes to understanding these associations over time, I do not believe that research in this area is original because multiple results have been achieved to date. However, this study does address a specific gap in the field. The strength of this study is the point in time when the long-term course was followed. The impact of the long-term course of the study was evaluated in comparison with the results of previous studies. As for methodology, I think the current cohort is unlikely to improve any further. The conclusions are consistent with the evidence and arguments presented. The references used are appropriate. I only have one concern, please see the detail comments.

Response: We thank the reviewer for careful reading and description about our manuscript with the valuable comments. We worked to the best of our abilities to deal with the issues reviewer point out. We thought that our study was differentiated compared to previous studies with regard to investigating whether n-3 PUFA status modify the effect of dietary antioxidants on depression. To clarify this, we described these main findings more specifically in the discussion part as follows.

- line 86-99

Our study showed that in the group with high n-3 PUFA intake, significant inverse correlation was found between CDAI and BDI scores among women. As discussed above, n-3 PUFAs can exert the potential anti-inflammatory effects via preventing or decreasing the inflammatory status. Also, numerous studies have demonstrated that a protective effect of consuming a diet rich in antioxidants on the risk of depression (56, 57). In experimental animals, combined treatment of n-3 PUFA and ascorbic acid provided an additive effect in suppressing lipid peroxidation compared to n-3 PUFA or ascorbic acid alone (58). A recent meta-analysis of human, which pooled data from 10 randomized controlled trials of n-3 fatty acids in patients with acute inflammatory lung injury, suggested that the enteral formulation which provided n-3 fatty acids in combination with antioxidant and γ-linolenic acid formulation led to more favorable outcomes (17). Regarding the results, it might be suggested that n-3 PUFA could exert more beneficial effects against depression, which is associated with inflammatory conditions, if accompanied by increased consumption of foods rich in antioxidants.

Detail comments: The results of this study suggest that economic status, such as job and annual income, may affect depression. The possibility that a higher annual income may result in different dietary habits should be considered.

Response: We truly appreciate with the reviewer's comments. As the reviewer stated, the socioeconomic factors might influence dietary habits. Considering the possibility that socioeconomic factors might result in different dietary habit, thus affecting the risk of depression, we adjusted for these potential confounders in the multivariable model with stepwise regression procedures. We added related contents in the discussion part with additional references as follows.

- line 8-19

In our study, socioeconomic status such as higher levels of education and income was associated with greater consumption of seafood. A systematic review of European countries revealed that the socioeconomic factors might influence dietary habits (33). A study analyzing data of 5,721 participants from the Korea National Health and Nutrition Examination Survey, reported that higher socioeconomic status was associated with the “modified” dietary pattern, which reflected good nutritional status (34). A cross-sectional study delineating yearly trends in the daily consumption of seafood and investigating the socioeconomic factors influencing seafood consumption among elderly Koreans, showed that there was a significant correlation between seafood intake and educational level and family income (35). Considering the possibility that socioeconomic factors might result in different dietary habit, thus affecting the risk of depression, we adjusted for these potential confounders.

Round 2

Reviewer 2 Report

The revised version has been much improved.

The revised version has been much improved.